# Development and Skin Penetration Pathway Evaluation Using Confocal Laser Scanning Microscopy of Microemulsions for Dermal Delivery Enhancement of Finasteride

**DOI:** 10.3390/pharmaceutics14122784

**Published:** 2022-12-13

**Authors:** Thirapit Subongkot, Natthan Charernsriwilaiwat, Rattathammanoon Chanasongkram, Kantawat Rittem, Tanasait Ngawhirunpat, Praneet Opanasopit

**Affiliations:** 1Research Unit of Pharmaceutical Innovations of Natural Products Unit (PhInNat), Faculty of Pharmaceutical Sciences, Burapha University, Saen Suk, Mueang, Chonburi 20131, Thailand; 2Yainthai Company Limited, Nonthaburi 11150, Thailand; 3Department of Pharmacognosy and Pharmaceutical Chemistry, Faculty of Pharmaceutical Sciences, Burapha University, Saen Suk, Mueang, Chonburi 20131, Thailand; 4Department of Industrial Pharmacy, Faculty of Pharmacy, Silpakorn University, Nakhon Pathom 73000, Thailand

**Keywords:** microemulsions, finasteride, poloxamer 124, skin penetration pathway, confocal laser scanning microscopy

## Abstract

This study aimed to develop microemulsions using poloxamer 124 as a surfactant to improve the skin penetration of finasteride and to investigate the skin penetration pathways of these microemulsions by colocalization techniques using confocal laser scanning microscopy (CLSM). The prepared finasteride-loaded microemulsions had average particle sizes ranging from 80.09 to 136.97 nm with particle size distributions within acceptable ranges and exhibited negative surface charges. The obtained microemulsions could significantly increase the skin penetration of finasteride compared to a finasteride solution. According to the skin penetration pathway evaluation conducted with CLSM, the microemulsions were hair follicle-targeted formulations due to penetration via the transfollicular pathway as a major skin penetration pathway. Additionally, this study found that the microemulsions also penetrated via the intercluster pathway more than via the intercellular pathway and transcellular pathway. The intercluster pathway, intercellular pathway, and transcellular pathway were considered only minor pathways.

## 1. Introduction

Androgenetic alopecia (AGA) is genetic hair loss observed in both males and females that is caused by an excessive response to androgens (male sex hormones) [1]. In men, AGA is characterized by an expansion of hair thinning from the frontal scalp to the vertex [1]. In women, it is characterized by diffuse hair thinning in the frontal and parietal scalp [2]. Although AGA has a high prevalence in elderly individuals, it is also found in puberty [3]. Hair thinning until visible scalp appears is perceived as an older appearance, which affects patient self-esteem and results in psychosocial morbidity [4]. In the scalp follicles of genetically susceptible men, androgens can suppress hair growth and stimulate hair follicle miniaturization, which finally leads to baldness [5]. In the blood circulatory system, testosterone is an important androgen. Testosterone circulates to the skin by blood capillaries and is converted by 5alpha-reductase within hair follicles to the more potent androgen, 5alpha-dihydrotestosterone (DHT) [6]. In a male mouse model, DHT inhibited hair regrowth, shortened the anagen phase, and induced hair miniaturization [7]. DHT also induced IL-6, leading to inhibition of hair shaft elongation [8].

Finasteride is the only US FDA-approved drug for the treatment of male AGA and is used at a dose of 1 mg orally once daily [9]. Finasteride acts as a 5alpha-reductase inhibitor preventing conversion of testosterone to DHT. According to progressive hair loss due to AGA, patients must take finasteride at an early stage and be treated for a long duration to achieve the best results. However, finasteride oral administration (5 mg per day) for 7 years causes serious side effects, such as the increased risk of high-grade prostate cancer and sexual side effects, such as decreased libido, erectile dysfunction, and ejaculatory disorders [10]. To reduce the adverse systemic effects due to long-term use of this drug, developing a topical treatment by targeting hair follicle delivery is necessary.

Dermal drug delivery consists of the transport of drugs or active substances to various skin layers. However, the stratum corneum, the outermost skin layer, is the rate-limiting layer in percutaneous absorption. The desirable physicochemical properties of drugs with good percutaneous absorption are low molecular weights (less than 500 daltons) and optimum log partition coefficients (log P) between 1 and 3 [11]. Finasteride is a lipophilic drug that has molecular weight and log *p* values of 372.54 daltons and 3.97, respectively [12]. Thus, finasteride exhibits poor dermal absorption.

The clinical study of 50 male AGA patients who were treated with topical finasteride at a concentration of 0.5% (approximately 5 mg/mL) twice daily showed significantly higher hair density than the placebo group [13]. There were several studies that prepared finasteride in liposomes [14,15]. Due to the highly lipophilic properties of finasteride, finasteride was encapsulated with a very small amount in liposomes (approximately 0.2 mg/mL), which might not provide enough concentration to achieve therapeutic efficacy. Therefore, the selection and development of a nanoparticle drug delivery system which could increase the solubility of finasteride for dermal delivery is essential.

Microemulsions are transparent colloidal drug delivery systems that consist of oil, surfactant, cosurfactant, and water. Microemulsions are thermodynamically stable systems with nanoscale-size particles. Microemulsions have many advantages, such as ease of preparation and the ability to incorporate both hydrophilic and lipophilic drugs. There have been many reports of microemulsions being used as dermal delivery systems for drugs such as tacrolimus [16], sertaconazole [17], caffeine [18], celecoxib [19], and minoxidil [20]. Because microemulsion formulation contains a high amount of surfactant which can improve the solubility of lipophilic drugs, microemulsions were selected as a delivery system for this study to increase the solubility and to enhance the dermal delivery of finasteride. This study also developed a novel microemulsion formulation as a delivery system for finasteride using poloxamer 124 as a surfactant, which has never been reported before. Poloxamer 124 is a nonionic polymeric surfactant that has a hydrophilic–lipophilic balance (HLB) between 12 and 18. The physicochemical properties of poloxamers differ from those of general surfactants, which can affect the skin penetration ability of the prepared microemulsions. This study also investigated the skin penetration pathways for the obtained microemulsions by labeling the particles with fluorescent dyes and conducted observations with confocal laser scanning microscopy (CLSM).

This study aims to develop microemulsions using a polymeric surfactant to increase dermal delivery of finasteride by targeting hair follicle transport and to evaluate the skin penetration pathways of these microemulsions.

## 2. Materials and Methods

### 2.1. Materials

Finasteride was purchased from Hunan Yuxin Pharmaceutical Co., Ltd., Hunan, China. Poloxamer 124 (Synperonic PE/L 44) was purchased from Croda Europe Limited, East Yorkshire, UK. Diethylene glycol monoethyl ether (Transcutol P) was purchased from Carlo Ebra, Milan, Italy. PEG-12 (Pluracare E 600) and PEG-35 castor oil (Kolliphor EL) were purchased from BASF Personal Care and Nutrition GmbH, Monheim, Germany. PEG-6 Caprylic/Capric Glycerides (Cetiol 767) were purchased from BASF Personal Care and Nutrition GmbH, Monheim, Germany. PEG-80 sorbitan laurate was purchased from Chanjao Longevity Co., Ltd., Bangkok, Thailand. Oleic acid, rhodamine B base, and finasteride (used as reference standards) were purchased from Sigma Aldrich, MO, USA. Medium-chain triglycerides (Lexol GT865) were purchased from Inolex, PA, USA.

4′,6-Diamidino-2-phenylindole, dihydrochloride (DAPI) and 1,2-dihexadecanoyl-sn-glycero-3-phosphoethanolamine triethylammonium salt (NBD-PE) were purchased from Thermo Fisher Scientific, Waltham, MA, USA. All other reagents were of analytical reagent grade and are commercially available.

### 2.2. Solubility Study

The solubility of finasteride was determined in various oils (e.g., medium-chain triglycerides, isopropyl myristate and oleic acid); surfactants (Poloxamer 124); and cosurfactants (e.g., Transcutol P, Tween 20, PEG-80 sorbitan laurate, PEG-35 castor oil, PEG-6 Caprylic/Capric Glycerides, and PEG-12) to select components with the highest solubility for microemulsion preparation. Finasteride was weighed excessively into glass containers with screw caps containing 2 mL of various solvents. The suspensions were stirred with a magnetic stirring bar for 24 h at room temperature. Then, the suspensions were transferred to microcentrifuge tubes and centrifuged at 10,000 rpm 25 °C for 15 min (Sorvall^TM^Legend^TM^ X1R Centrifuge, Thermo Scientific^TM^, Waltham, MA, USA). The obtained supernatants were diluted with appropriate amounts of methanol and quantitatively analyzed by high-performance liquid chromatography (HPLC). The oil and cosurfactant that provided the highest solubility of finasteride were selected as the oil phase and cosurfactant for microemulsion preparation.

### 2.3. Preparation of Microemulsions

#### 2.3.1. Construction of a Pseudoternary Phase Diagram of Microemulsions

A pseudoternary phase diagram of the microemulsions was devised to determine the microemulsion region. Microemulsions consisting of oil, poloxamer 124 as the surfactant, cosurfactant, and ultrapure water as the water phase were prepared using the water titration method. Poloxamer 124 was mixed with the cosurfactant to provide a surfactant mixture (Sm). Both the oil phase and Sm were accurately weighed and mixed together at various ratios of 0.25:4.75, 0.5:4.5, 1:4, 1.5:3.5, and 2:3 *w*/*w*. These oil and Sm mixtures were stirred with a magnetic stir bar. Then, water was added dropwise into the oil and Sm mixtures until they changed from transparent to translucent and finally became turbid. Microemulsions were defined as transparent mixtures. After the pseudoternary phase diagram of the microemulsions was obtained, each concentration inside the microemulsion region was prepared as a blank microemulsion and checked for particle size using the photon correlation spectroscopy (PCS) technique as described in Section 2.4. Blank microemulsions with average particle sizes not greater than 100 µm were selected to prepare the finasteride-loaded microemulsions.

#### 2.3.2. Preparation of Finasteride-Loaded Microemulsions

Rossi et al. [21] reported that a combined treatment using 0.5% topical finasteride and 2% minoxidil was efficacious in treating postmenopausal female pattern hair loss and provided better results than 0.05% 17α-estradiol with 2% minoxidil. This study, therefore, prepared a finasteride-loaded microemulsion at a concentration of 0.5% *w*/*v*.

Blank microemulsions were prepared from the microemulsion region within the pseudoternary phase diagram by mixing oil, Sm, and water and stirring with a magnetic bar until homogeneous mixtures were obtained. Finasteride was weighed into volumetric flasks filled with blank microemulsions before being placed in a sonicator bath until clear mixtures were obtained.

### 2.4. Characterization of Microemulsions: Mean Particle Size, Surface Charge, Particle Size Distribution, and Electrical Conductivity

The average particle size, surface charge (zeta potential), particle size distribution or polydispersity index (PDI), and conductivity for each microemulsion formulation were determined by filling the microemulsions in folded capillary cells without dilution using the photon correlation spectroscopy (PCS) technique of a particle size analyzer (Zetasizer Nano-ZS, Malvern Instrument, Worcestershire, UK) equipped with a 4 mW He–Ne laser at a scattering angle of 173°.

### 2.5. In Vitro Skin Penetration Study

#### 2.5.1. Preparation of Skin

Skin samples from the abdominal parts of naturally dead neonatal pigs obtained from Charnchai Farm (Ratchaburi, Thailand) were used as barrier membranes for skin penetration tests. The muscles that might have been attached to the skin during cutting and peeling skin from the body were carefully removed with a surgical blade. The obtained skin samples consisted of epidermis, dermis, and subcutaneous tissue with thicknesses not more than 1 mm. Prior to the experiment, the skin samples, which were stored in a refrigerator at −40 °C, were thawed and washed with phosphate-buffered saline (PBS) before use.

#### 2.5.2. Skin Penetration Tests

The control formulation used in this test was a 0.5% *w*/*v* finasteride solution in which the drug was dissolved in a solvent composed of ethanol:propylene glycol:water at a ratio of 32.3:32.3:35.4 *w*/*w*/*w*. Skin penetration tests of each formulation were performed using water-jacketed Franz diffusion cells connected to a circulating bath to control the temperature at 32 °C. PBS was filled in the receiver compartment with a volume of 6.5 mL and stirred gently with a magnetic stirring bar. The skin obtained, as described in Section 2.5.1, was mounted between the donor and receiver compartments by orienting the epidermis to face the donor compartment. A total of 2 mL of each formulation was added to the donor compartment. The donor compartment and sampling port were covered with Parafilm^®^ throughout the experiment to prevent evaporation. After treatment for 6 h, the receiver medium was withdrawn for drug concentration analysis with HPLC. The formulation in the donor part was removed, and the skin sample was washed with PBS twice to remove any excess drug. The drug quantities in the treated skin samples were analyzed by the tape strip method as described by Subongkot and Sirirak [19]. The drug that remained on the skin was wiped away once by tissue paper. The treated skin sample was placed on a dissection tray with wax and fixed by a pin. The stratum corneum layers were removed from the tissue using 24 mm wide adhesive tape (Scotch^®^ Transparent Tape 500, 3M, Bangkok, Thailand) 35 times. These tape strips, except for the first tape, were immersed in a tightly closed glass vial containing 5 mL of methanol and sonicated for 15 min in a sonicator bath. Then, 1 mL of the extracted methanol was pipetted into a microcentrifuge tube and centrifuged at 10,000 rpm and 25 °C for 15 min. The amount of finasteride in the obtained supernatant was analyzed by HPLC. The drug quantity that permeated into the stratum corneum was calculated from the following Equation (1):Drug amount in the stratum corneum (µg/cm^2^) = Qs/P(1)
Qs = finasteride quantity in the stratum corneum (µg)
P = skin penetration area (cm^2^)

The remaining skin from each sample was cut into small pieces and placed into a tightly closed glass vial containing 3 mL of methanol. The vial was placed in a sonicator bath for 15 min. After that, the extracted methanol was centrifuged at 10,000 rpm and 25 °C for 15 min. The finasteride quantity in the supernatant was determined by HPLC. The finasteride quantity in the viable epidermis (VE), dermis, and subcutaneous tissue was calculated from the following Equation (2):Finasteride quantity in the VE, dermis, and subcutaneous tissue (µg/cm^2^) = Qvd/P(2)
Qvd = finasteride quantity in the VE, dermis, and subcutaneous tissue (µg)
P = skin penetration area (cm^2^)

The following equation was used to calculate the enhancement ratio (Equation (3)):Enhancement ratio (ER) = finasteride quantity in the VE, dermis, and subcutaneous tissue from finasteride-loaded microemulsions/drug quantity in the VE, dermis, and subcutaneous tissue from the finasteride solution(3)

### 2.6. Skin Penetration Pathway Evaluation

The microemulsion formulation that provided the highest level of finasteride penetration into the viable epidermis (VE), dermis, and subcutaneous tissue was selected to investigate its skin penetration pathways by visualization with CLSM using a colocalization technique. Rhodamine B base, a lipophilic fluorescent dye with log *p* = 1.95 [22], was used as the entrapped drug because it has a log *p* value similar to finasteride. The microemulsion particles were probed with NBD-PE, a phospholipid surfactant conjugated with a green fluorophore. Thus, the entrapped drug exhibited red fluorescence, whereas the microemulsion particles exhibited green fluorescence.

#### 2.6.1. Preparation of Rhodamine B Base-Loaded NBD-PE Probed Microemulsions and Rhodamine B Base Solution

To prepare the rhodamine B base-loaded NBD-PE probed microemulsions, 11.62 mg of rhodamine B base and 5 mg of NBD-PE were weighed into volumetric flasks. Then, the volumes were adjusted with blank microemulsion to 2 mL. To prepare the rhodamine B base solution, 29.06 mg of rhodamine B base was weighed, and the volume was adjusted to 5 mL with a mixture of ethanol:propylene glycol:water at a ratio of 32.3:32.3:35.4 *w*/*w*/*w*. Both the rhodamine B base-loaded NBD-PE probed microemulsions and rhodamine B base solution had the same rhodamine B base concentration of 5.8 mg/mL.

#### 2.6.2. Skin Penetration Test

Skin penetration tests of the rhodamine B base-loaded NBD-PE probed microemulsions and rhodamine B base solution were performed using Franz diffusion cells as described in Section 2.5.2. The rhodamine B base-loaded NBD-PE-conjugated microemulsions were pipetted (400 µL) into the donor parts of 3 diffusion cells and covered with Parafilm^®^. After treatment for 30 min, 1 h, and 2 h, each formulation was removed, and the skin was washed with PBS to remove excess dye. For the case of the rhodamine B base solution treatment, 400 µL of solution was pipetted into the donor part and treated for 1 h. After treatment for 1 h, the formulation in the donor part was removed, and the skin was washed with PBS to remove excess dye. A portion of the skin was separated for the tissue cross-section study.

#### 2.6.3. Preparation of Tissue Cross-Section Procedures

The treated skin described in Section 2.6.2 was cross-sectioned with a cryomicrotome (Leica 1850, Leica Instruments GmbH, Nussloch, Germany). Tissue was frozen at −30 °C for 10 min in a cryomicrotome, which was precooled at −30 °C. Then, the frozen tissue was placed into a sample holder and embedded with optimal cutting temperature compound (Bio-Optica, Milan, Italy). The embedded tissue was sectioned at a thickness of 5 µm onto a positively charged microscope slide (Bio-Optica, Milan, Italy). The sectioned specimen was stained for nucleic acids with 10 µg/mL DAPI solution for 15 s to determine the living cell layers in the skin. Then, the microscope slide was immersed into a beaker containing water to remove excess dye. The microscope slide was allowed to dry at room temperature, mounted immediately with synthetic resin mounting medium, and covered with a coverslip.

#### 2.6.4. CLSM Visualization

A CLSM study of whole skin and sectioned tissue was performed with an inverted Zeiss LSM 800 with Airyscan (Carl Zeiss Microscopy GmbH, Jena, Germany) using a 10× objective lens. To visualize the fluorescence localizations of the rhodamine B base-loaded NBD-PE probed microemulsions on the skin surface, treated skin was placed onto a 22 mm × 50 mm coverslip (MENZEL-GLÄSER^®^, Braunschweig, Germany) using a small volume of PBS as an immersion medium by orienting the epidermis toward the objective lens (EC Plan Neofluar 10×), which was equipped with four diode lasers. Green fluorescence (dye name AF488) was detected at excitation and emission wavelengths of 493 and 517 nm, respectively. Red fluorescence (dye name AF568) was detected at excitation and emission wavelengths of 577 and 603 nm, respectively. The sectioned tissue was observed using the same objective lens, for which the blue fluorescence from DAPI staining was detected at excitation and emission wavelengths of 353 and 465 nm, respectively. The fluorescence intensities of the image were analyzed by Zen software (Blue edition, Carl Zeiss Microscopy GmbH, Jena, Germany).

### 2.7. HPLC Analysis

Finasteride quantities were quantitatively analyzed by a validated method using HPLC (Shimadzu, Kyoto, Japan). The samples were injected into a C18 reversed-phase column with a particle size of 5 µm, which had dimensions of 4.6 mm × 150 mm (GL Sciences, Tokyo, Japan). The mobile phase consisted of acetonitrile:water at a ratio of 70:30 *v*/*v* using a flow rate of 1 mL/min. The sample injection volume was 20 µL, and the UV detection wavelength was 210 nm. A quantitative analysis of finasteride was obtained from the linear standard curve (R^2^ = 0.9999) in a concentration range of 0.5–200 µg/mL. The limit of detection and limit of quantitation were 0.57 and 1.73 µg/mL, respectively. For the accuracy and precision, the accuracy was expressed as the percentage of recovery, and the precision was expressed as the percentage coefficient of variation (CV). For this analysis, the percentage recovery was 100.28 ± 0.89, and the percentage CV was 1.85.

### 2.8. Statistical Analysis

All data were statistically analyzed by using analysis of variance (ANOVA), followed by a post hoc test (LSD). Differences of *p* < 0.05 were considered statistically significant.

## 3. Results and Discussion

### 3.1. Solubility Study

The solubility data for finasteride in the different oils, surfactants, and cosurfactants are shown in Table 1. Among the different oils, oleic acid provided the highest finasteride solubility. Therefore, oleic acid was selected as the oil phase for preparing the microemulsions. Among the different cosurfactants, finasteride had the highest solubility in Transcutol P. Thus, Transcutol P was used as a cosurfactant for preparing the microemulsions.

### 3.2. Preparation of Microemulsions: Construction of a Pseudoternary Phase Diagram of Microemulsions and Preparation of Finasteride-Loaded Microemulsions

A pseudoternary phase diagram of microemulsions consisting of oleic acid as the oil phase, poloxamer 124 as the surfactant, Transcutol P as the cosurfactant, and ultrapure water as the aqueous phase is shown in Figure 1. Different blank microemulsions were randomly prepared using various concentrations that fell within the microemulsion region. The sizes of the selected blank microemulsion formulations were measured with the PCS technique. The criterion for selecting the blank microemulsion formulations for loading with finasteride was a particle size not more than 100 µm. Three blank microemulsion formulations with particle sizes that were consistent with the above criterion are shown in Table 2. According to a clinical study of 119 female patients with female pattern hair loss, topical 0.5% finasteride with 2% minoxidil showed higher treatment efficacy than 0.05% 17 alpha estradiol with 2% minoxidil. Therefore, the finasteride concentration used was 0.5% *w*/*v*. Finasteride was dissolved in three blank microemulsions (e.g., P1, P2, and P3) to prepare finasteride-loaded microemulsions with concentrations of 0.5% *w*/*v*. The obtained 0.5% *w*/*v* finasteride-loaded microemulsions had transparent properties, which indicated that finasteride could be completely dissolved in these microemulsion formulations. There was a report of microemulsions composed of cinnamal oil as the oil phase, Tween 20 and propylene glycol as the surfactant and cosurfactant, respectively, and water as the aqueous phase. These reported microemulsions could be used to prepare finasteride microemulsions at a concentration of 0.3% [23]. However, this study prepared finasteride microemulsions up to a concentration of 0.5%.

### 3.3. Characterization of Microemulsions: Mean Particle Size, Surface Charge, Particle Size Distribution, and Electrical Conductivity

The mean particle sizes, zeta potentials, PDIs, and conductivities of the blank microemulsions and finasteride-loaded microemulsions are shown in Table 3. The mean particle sizes of the blank microemulsions ranged from 51.18 to 66.12 nm, while the particle sizes of the finasteride-loaded microemulsions ranged from 80.09 to 136.97 nm. There were only finasteride-loaded microemulsions (P2), and the mean particle size did not differ from that of blank microemulsions (*p* = 0.351). The mean PDIs of the blank microemulsions were in the range of 0.15–0.39, while those of the finasteride-loaded microemulsions were in the range of 0.21–0.37. The prepared microemulsions had acceptable PDI ranges. The zeta potentials of the blank microemulsions ranged from −0.34 to −0.2 mV, whereas the zeta potentials of the finasteride-loaded microemulsions ranged from −0.47 to −0.11 mV. Regarding the components of the microemulsions, oleic acid had a negative surface charge due to the carboxylic group. Poloxamer 124, Transcutol P, and finasteride are nonionic molecules. Thus, finasteride-loaded microemulsions exhibited negative charges due to the oleic acid. The conductivities of the blank microemulsions ranged from 57.83 to 77.23 µS/cm, while the conductivities of the finasteride-loaded microemulsions ranged from 57.97 to 72.97 µS/cm. Microemulsion types can be classified as oil-in-water microemulsions, water-in-oil microemulsions, and bicontinuous phase microemulsions [24,25]. The electrical conductivities of microemulsions can be used to determine microemulsion types. Krauel et al. [26] prepared different microemulsion types consisting of ethyl oleate as the oil phase, sorbitan monolaurate as the surfactant, polyoxyethylene 20 sorbitan mono-oleate as the cosurfactant, and water as the aqueous phase. Microemulsions with conductivities less than 1 µS/cm were categorized as water-in-oil microemulsions, whereas microemulsions with conductivities between 1 and 10 µS/cm were categorized as bicontinuous phase microemulsions. Microemulsions with conductivities greater than 10 µS/cm were categorized as oil-in-water microemulsions. This study, therefore, suggested that all blank microemulsions and all finasteride-loaded microemulsions (e.g., P1, P2, and P3) were oil-in-water microemulsions.

### 3.4. In Vitro Skin Penetration Study

The finasteride quantities in the control (finasteride solution) and the different microemulsions that penetrated into different skin layers, which were the stratum corneum, viable epidermis, dermis, subcutaneous tissue, and receiver medium, are shown in Table 4.

The goal of developing this delivery system is to deliver finasteride into hair follicles to inhibit the 5alpha-reductase enzyme. Because hair follicles are located in the epidermis and dermis, improvements in the skin penetration efficiency are evaluated based on the drug quantities that permeated into the viable epidermis, dermis, and subcutaneous tissues. The finasteride amounts that penetrated the viable epidermis, dermis, and subcutaneous tissue from microemulsions (P2) were significantly higher than those of the control (*p* = 0.001). The finasteride quantities that penetrated the viable epidermis, dermis, and subcutaneous tissue from P2 were significantly higher than those of P1 (*p* < 0.001) and P3 (*p* = 0.001). The finasteride quantities that penetrated the viable epidermis, dermis, and subcutaneous tissue from P1 were significantly higher than those of the control (*p* = 0.001). The finasteride quantities that penetrated the viable epidermis, dermis, and subcutaneous tissue from P3 were significantly higher than those of the control (*p* = 0.011). The finasteride quantities that penetrated the viable epidermis, dermis, and subcutaneous tissue from P1 did not differ from those of P3 (*p* = 0.122). In summary, all microemulsion formulations could increase skin penetration of finasteride more than the finasteride solution formulation. Among the microemulsion formulations, P2 provided the highest finasteride quantities to the viable epidermis, dermis, and subcutaneous tissue. Therefore, P2 was chosen to further investigate of the skin penetration pathways and mechanisms of action. Among the microemulsion formulations, P2 contained the highest amount of oleic acid and had the smallest mean particle size. It is suggested that the two mentioned factors were the major mechanisms that affected the increased skin penetration of finasteride when compared to the other formulations.

The stratum corneum is the outermost skin layer, which provides a barrier to chemical penetration and microorganism invasion. The stratum corneum consists of dead keratinocytes or corneocytes embedded in an intercellular lipid matrix that is similar to the arrangement of brick and mortar structures. Keratin is present inside corneocytes, whereas the major components in intercellular lipids consist of ceramides, cholesterol, and free fatty acids [27].

Francoeur et al. [28] investigated the effect of oleic acid on the stratum corneum of pig skin using differential scanning calorimetry (DSC). The authors suggested that oleic acid increased the fluidity of intercellular lipids, resulting in an enhancement of skin penetration by drugs. Rowat et al. [29] evaluated the mechanisms of skin penetration enhancement of oleic acid with a stratum corneum model membrane composed of ceramide, cholesterol, and palmitic acid. The authors concluded that oleic acid increased skin penetration by interacting and extracting endogenous stratum corneum membrane components. This interaction led to a reduction in the crystallinity of intercellular lipids and creation of a more permeable oleic acid-rich region.

Poloxamer 124 and Transcutol P are nonionic surfactants that can increase skin penetration by denaturing the keratin inside corneocytes [30]. Surfactants also partition into the intercellular lipids of the stratum corneum to increase the fluidity of intercellular lipids [31].

It is generally accepted that particle size plays an important role in the efficacy of skin penetration. Su et al. [32] prepared nanoemulsions with different particle sizes, namely, 80, 200, and 500 nm. The formulation with the smallest particle size could more deeply penetrate the skin via hair follicles than those formulations with larger particle sizes. In this study, finasteride-loaded P2, which provided the highest amount of drug into the viable epidermis, dermis, and subcutaneous tissues, had the smallest particle size (80.09 nm) among the microemulsion formulations. Thus, these results also confirmed that small particle sizes could increase skin penetration via hair follicles better than larger particle sizes.

The mechanisms that increase the skin penetration efficiency of finasteride from microemulsions (P2) might result from both microemulsion components as penetration enhancers and small particle sizes that facilitate hair follicle penetration as described above.

### 3.5. Skin Penetration Pathway Evaluation

Theoretically, three skin penetration pathways have been proposed: the intercellular pathway, transcellular pathway, and transfollicular pathway [33]. The intercellular pathway or intercorneocyte pathway is represented by drug diffusion between adjacent corneocytes, while the transcellular pathway is represented by diffusion across corneocytes. The transfollicular pathway consists of transport via hair follicles by bypassing the stratum corneum barrier to the dermis. To investigate the skin penetration pathways of these novel microemulsions, a colocalization technique using multifluorescence-labeled particles was applied and visualized with CLSM. Rhodamine B base, which exhibits red fluorescence, was used as the entrapped drug, whereas NBD-PE, which exhibits green fluorescence, was used to label the microemulsion particles.

CLSM images of pig skin samples treated with rhodamine B base-loaded NBD-PE probed particles from different areas are shown in Figure 2. The red fluorescence of rhodamine B base and green fluorescence of NBD-PE were clearly deposited at the hair follicles more than in the nonfollicular region, indicating that the microemulsion particles penetrated the skin via hair follicles as a major penetration pathway.

In addition to the intercellular pathway, transcellular pathway, and transfollicular pathway, an intercluster pathway was discovered that was first reported in nude mouse skin by Schätzlein and Cevc [34]. When the skin was observed with microscopy, small wrinkles appeared on the epidermis that separated the clusters of corneocytes by gorges or canyons. The intercluster pathway was defined as the canyons between clusters of corneocytes, with depths beginning from the stratum corneum to the stratum basale. Therefore, intercellular and transcellular pathways were classified as extracluster pathways. According to the skin penetration study of transfersome, which was labeled with red fluorescence (1,2-dihexadecanoyl-sn-glycero-3-phosphoethanolamine-*N*-Lissamine^TM^ rhodamine B sulfonyl, triethylammonium salt, Rh-PE) and observed with CLSM, these particles penetrated via the intercluster pathway more than via the intercellular pathway [31].

Carrer et al. [35] studied the effects of different types of liposomes labeled with Rh-PE and 6-dodecanoyl-2-dimethylaminonaphthalene (Laurdan) on pig skin penetration by observations using multiphoton microscopy. The authors found that the structure of the intercluster region was physically related to the stratum corneum, which the lipophilic compound (Rh-PE) penetrated via the intercluster pathway more than via the extracluster pathway (intercellular or transcellular pathway). The penetration of Rh-PE in the intercluster region was observed to occur deeply into the layers of the stratum basale. However, the authors suggested that the intercluster pathway might act as a lipophilic drug reservoir.

CLSM images of the intercluster pathway of pig skin treated with rhodamine B base-loaded NBD-PE probed microemulsions are shown in Figure 3. The fluorescence intensities were analyzed by covering the area between the extracluster pathway and intercluster pathway of the skin, as shown in Figure 3e,f. The fluorescence intensities of the red and green fluorescence of the intercluster pathway were higher than those of the extracluster pathway, indicating that rhodamine B base-loaded NBD-PE probed microemulsions penetrated via the intercluster pathway more than via the extracluster pathway. This study suggested that microemulsions penetrated the skin via the intercluster pathway more than via the intercellular and transcellular pathways.

CLSM images of cross-sectioned pig skin treated with rhodamine B base solution at 1 h are shown in Figure 4. Structurally, the stratum corneum of the treated skin was still intact, as seen in Figure 4c. The red fluorescence of rhodamine B base was mainly located at the stratum corneum more than at the hair follicles, indicating that the solution formulation penetrated through the intercellular and transcellular pathways more than via the transfollicular pathway.

CLSM images of cross-sectioned pig skin treated with rhodamine B base-loaded NBD-PE probed microemulsions at 30 min, 1 h, and 2 h are shown in Figure 5a–c, respectively. At 30 min, there was red and green fluorescence at the hair follicles and stratum corneum, and the stratum corneum was intact. At 1 h, red and green fluorescence was found in the stratum corneum, viable epidermis, dermis, subcutaneous tissue, and hair follicles that was stronger than at 30 min. However, the stratum corneum was found to be detached more than at 30 min. Stronger red and green fluorescence was located at the hair follicle region than at the nonfollicular region, indicating that the microemulsions penetrated via the transfollicular pathway as a major skin penetration pathway. Although the microemulsions penetrated via the intercluster pathway more than via the intercellular and transcellular pathways, as observed in Figure 3e and Figure 3f, the obtained cross-section images indicated that the microemulsions still permeated the skin via the transfollicular pathway more than via the intercluster pathway. This study suggested that the intercluster pathway acts as a lipophilic drug reservoir that cannot bypass the particle transport from the stratum corneum to the dermis better than the transfollicular pathway.

At 2 h, there was red and green fluorescence throughout the tissue. However, the stratum corneum was found to be detached after more than 1 h. The stratum corneum is composed of corneocytes that adhere together by corneodesmosomes [36]. This study, therefore, suggested that the detachment of the stratum corneum, as seen in Figure 5(b-4,c-4), resulted from the impact of the surfactants in the microemulsion components, which caused corneodesmosome degradation [37]. In addition, transport of particles through the transfollicular pathway was the main effect for the skin penetration enhancement of finasteride, and the denaturing mechanism of corneodesmosomes was responsible for the synergistic effects that improved skin penetration.

## 4. Conclusions

This study successfully prepared finasteride-loaded microemulsions using poloxamer 124 as a surfactant with a finasteride concentration of 0.5% *w*/*v*. The obtained microemulsions had nanoscale particle sizes with particle size distributions within acceptable ranges that exhibited negative surface charges. The microemulsions that provided the highest levels of skin penetration of finasteride had the smallest particle sizes compared to the other formulations. This study suggested that small particle sizes could facilitate delivery via hair follicles by bypassing the stratum corneum to the dermis more effectively than could larger particles. According to the skin penetration pathway evaluation of the microemulsions, it was suggested that the microemulsion particles penetrated via the transfollicular pathway as a major penetration pathway followed by the intercluster pathway, whereas the intercellular pathway and transcellular pathway were only minor pathways.

## Figures and Tables

**Figure 1 pharmaceutics-14-02784-f001:**
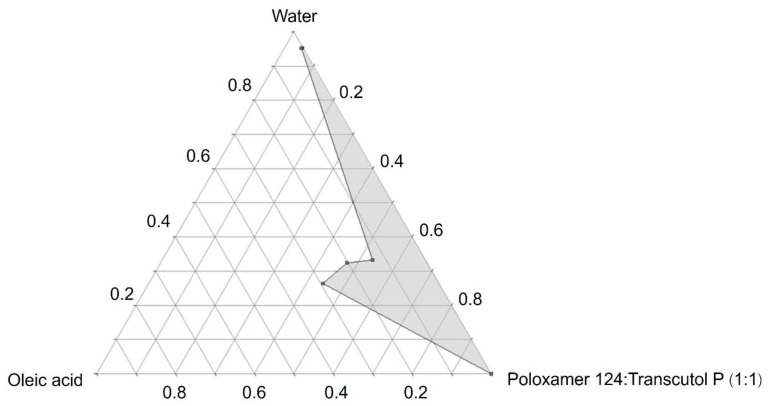
Pseudoternary phase diagram composed of oleic acid as the oil phase, poloxamer 124 as the surfactant, Transcutol P as the cosurfactant, and water as the aqueous phase. The surfactant:cosurfactant ratio = 1:1 *w*/*w*. The gray area represents the microemulsion region.

**Figure 2 pharmaceutics-14-02784-f002:**
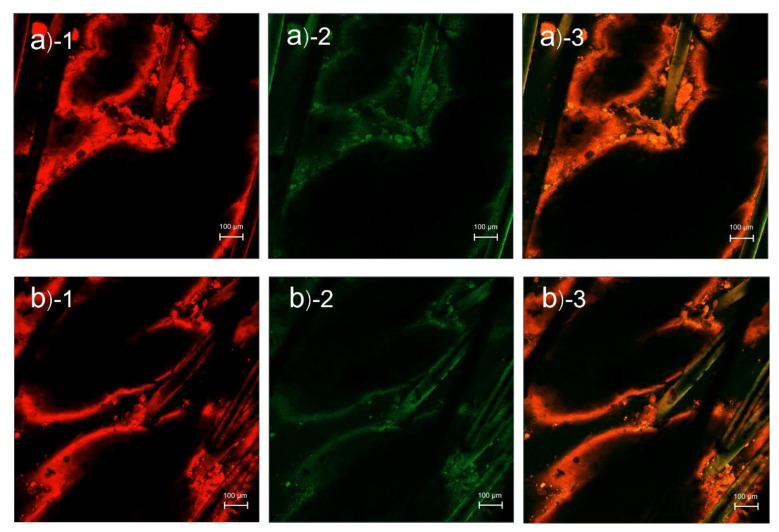
CLSM images (top view) of pig skin treated with rhodamine B base-loaded NBD-PE probed microemulsions from different areas: (**a**,**b**) at 2 h; (1) red fluorescence from rhodamine B base, (2) green fluorescence from NBD-PE, and (3) merged image of (1) and (2). The scale bar represents 100 µm.

**Figure 3 pharmaceutics-14-02784-f003:**
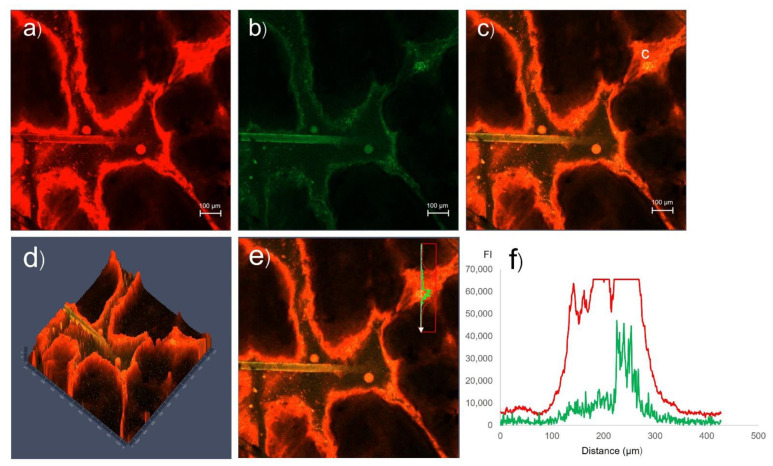
CLSM images of the intercluster pathway of pig skin treated with rhodamine B base-loaded NBD-PE probed microemulsions at 30 min: (**a**) red fluorescence from rhodamine B base; (**b**) green fluorescence from NBD-PE; and (**c**) merged image of (**a**) and (**b**) (c = canyons); (**d**) 3D image of (**c**); (**e**) region used for fluorescence intensity determination (white arrow); (**f**) fluorescence intensity profiles of red fluorescence (red line) and green fluorescence (green line) from (**e**). The scale bar represents 100 µm.

**Figure 4 pharmaceutics-14-02784-f004:**
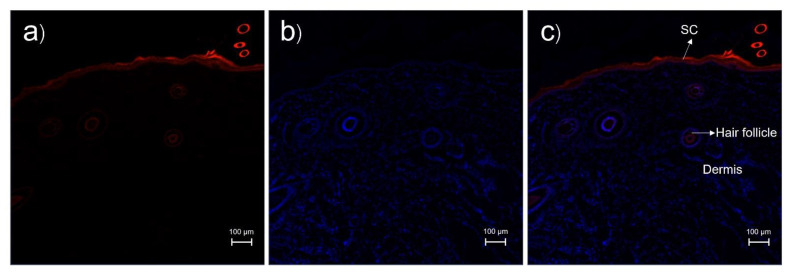
CLSM images of cross-sectioned pig skin treated with rhodamine B base solution and stained with DAPI at 1 h: (**a**) red fluorescence from rhodamine B base; (**b**) blue fluorescence from DAPI; (**c**) merged image of (**a**) and (**b**). The scale bar represents 100 µm. SC = stratum corneum.

**Figure 5 pharmaceutics-14-02784-f005:**
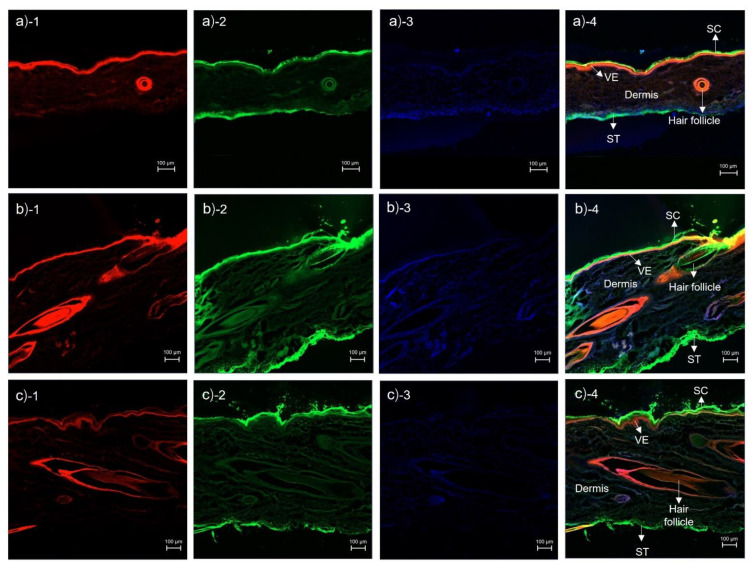
CLSM images of cross-sectioned pig skin treated with rhodamine B base-loaded NBD-PE probed microemulsions and stained with DAPI at (**a**) 30 min, (**b**) 1 h, and (**c**) 2 h. (1) Red fluorescence from rhodamine B base, (2) green fluorescence from NBD-PE, (3) blue fluorescence from DAPI, and (4) merged image of (1) and (2). The scale bar represents 100 µm. SC = stratum corneum, VE = viable epidermis, ST = subcutaneous tissue.

**Table 1 pharmaceutics-14-02784-t001:** Solubility of finasteride in various oils and surfactants.

Oils/Surfactants	Solubility (mg/mL)
Medium-chain triglycerides	2.31 ± 0.03
Isopropyl myristate	3.24 ± 0.61
Oleic acid	42.63 ± 0.04
Transcutol P	68.05 ± 0.07
Tween 20	17.59 ± 0.51
PEG-80 sorbitan laurate	18.48 ± 0.35
PEG-6 Caprylic/Capric Glycerides	12.25 ± 0.14
Kolliphor EL	15.59 ± 0.39
PEG-12	7.31 ± 0.04
Poloxamer 124	4.11 ± 0.16

Each value represents the mean ± standard deviation (n = 3).

**Table 2 pharmaceutics-14-02784-t002:** Formulations, component ratios, and concentrations of finasteride loaded in microemulsions.

Formulations	Ratio			Finasteride-Loaded Microemulsion (% *w*/*v*)
	Oleic Acid	Mixtures of Surfactants (Poloxamer 124:Transcutol P)	Water	
P1	15	60	25	0.5
P2	20	55	25	0.5
P3	10	70	20	0.5

**Table 3 pharmaceutics-14-02784-t003:** Particle sizes, zeta potentials, polydispersity indices (PDI), and electrical conductivities of microemulsions without finasteride (blank ME) and finasteride-loaded microemulsions (finasteride-loaded ME).

Formulations	Particle Size (nm)		Polydispersity Index (PDI)		Zeta Potential (mV)		Electrical Conductivity (µS/cm)	
	Blank MEs	Finasteride-Loaded MEs	Blank MEs	Finasteride-Loaded MEs	Blank MEs	Finasteride-Loaded MEs	Blank MEs	Finasteride-Loaded MEs
P1	51.18 ± 6.49	115.63 ± 24.52	0.39 ± 0.07	0.21 ± 0.01	−0.2 ± 0.1	−0.47 ± 0.11	77.23 ± 0.21	72.97 ± 2.71
P2	66.12 ± 1.79	80.09 ± 16.67	0.31 ± 0.05	0.37 ± 0.03	−0.24 ± 0.08	−0.37 ± 0.21	57.83 ± 0.15	59.8 ± 3.6
P3	57.66 ± 12.61	136.97 ± 27.94	0.15 ± 0.02	0.29 ± 0.03	−0.34 ± 0.29	−0.11 ± 0.02	55.93 ± 0.41	57.97 ± 0.41

Each value represents the mean ± standard deviation (n = 3).

**Table 4 pharmaceutics-14-02784-t004:** Amounts of finasteride that penetrated into the skin and receiver medium.

Formulations	Stratum Corneum (µg/cm^2^)	Viable Epidermis, Dermis, and Subcutaneous Tissue (µg/cm^2^)	Receiver Medium (µg/mL)	Enhancement Ratio
Solution	6.77 ± 5.38	1.88 ± 0.76	ND	-
P1	23.84 ± 15.75	5.17 ± 0.44	ND	2.75
P2	45.52 ± 25.29	10.55 ± 1.14 *	ND	5.61
P3	21.46 ± 1.56	4.03 ± 0.71	ND	2.14

Each value represents the mean ± standard deviation (n = 3). ND = not detectable. * *p* < 0.05 compared to solution.

## Data Availability

Not applicable.

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
