# Peer review of "Development and Skin Penetration Pathway Evaluation Using Confocal Laser Scanning Microscopy of Microemulsions for Dermal Delivery Enhancement of Finasteride"

_pharmaceutics, 2022, doi:10.3390/pharmaceutics14122784_

Round 1
Reviewer 1 Report
Comments and Suggestions for Authors
In the present manuscript, the authors reported the interesting study focusing on development and skin penetration pathway evaluation using confocal laser scanning microscopy of microemulsions for dermal delivery enhancement of finasteride
The manuscript is well written however, there are few comments that needs to be addressed.
The authors should pay more attention on the following points.
1. Line 46: “finasteride administration causes serious side effects” If it is administered at what dose(level), via what route and for how long? Please elaborate.
2. Line 60: “finasteride exhibits poor dermal absorption” rephrase this sentence. You are contradicting yourself because the Log P and molecular mass actually meet the requirements stated above.
3. What is the novelty of this study because finasteride has been delivered using various nanodelivery systems?. In your introduction briefly explain why you selected the microemulsion of your choice.
4. Line 70: between 12-18 what?
5. Line 121: How did you confirm the size of the microemulsion? Briefly state the method and the instrument utilized.

Reviewer 2 Report
The manuscript provides an extensive and timely focus work on Development and investigation of skin penetration pathway using confocal microscopy for topical delivery of microemulsions with finasteride. The article is well planned, However, due to poor quality of data presentation the article won’t be suitable to attract high attention with the readers of the MDPI-Pharmaceutics.
As such, I recommend acceptance after considering the suggestions as described below:
Ø The introduction section is too short and thus requires additional information related to prove the importance of work with androgenetic alopecia (AGA). So, I request authors to look carefully over the introduction part and modify it properly.
Ø Authors have claimed to use water jacketed Franz diffusion cells at 32 °C. I suggest authors to mention make and model of the Franz diffusion cells and believe the temperature might be 37±0.5° C. However, if permeability studies are performed at 32 °C please explain the significance of 32 °C.
Ø Please include a graphical/ schematic diagram which support your hypothesis
Ø I suggest authors to plan for histopathology studies and furnish some histopathological data to support their concept.
Reviewer 3 Report
The authors have done good work. But I can see their previous paper on same molecule and same formulation. No significant advantage can be seen in this work/no significant novelty. Only assessed with change in polymer. It could be advantageous if activity was determined and compared with previous formulation with cinnamon oil.
Rather previous work has done extensive screening using design expert.
Round 2
Reviewer 3 Report
I think authors have sufficiently defended novelty. The manuscript can be considered for publication